

# The state of OA: a large-scale analysis of the prevalence and impact of Open Access articles

Heather Piwowar[1,*], Jason Priem[1,*], Vincent Larivière[2,3], Juan Pablo Alperin[4,5], Lisa Matthias[6], Bree Norlander[7,8], Ashley Farley[7,8], Jevin West[7] and Stefanie Haustein[3,9]

[1] Impactstory, Sanford, NC, USA
[2] École de bibliothéconomie et des sciences de l'information, Université de Montréal, Montréal, QC, Canada
[3] Observatoire des Sciences et des Technologies (OST), Centre Interuniversitaire de Recherche sur la Science et la Technologie (CIRST), Université du Québec à Montréal, Montréal, QC, Canada
[4] Canadian Institute for Studies in Publishing, Simon Fraser University, Vancouver, BC, Canada
[5] Public Knowledge Project, Canada
[6] Scholarly Communications Lab, Simon Fraser University, Vancouver, Canada
[7] Information School, University of Washington, Seattle, USA
[8] FlourishOA, USA
[9] School of Information Studies, University of Ottawa, Ottawa, ON, Canada
[*] These authors contributed equally to this work.

Corresponding authors
Heather Piwowar,
heather@impactstory.org
Jason Priem, jason@impactstory.org

## ABSTRACT

Despite growing interest in Open Access (OA) to scholarly literature, there is an unmet need for large-scale, up-to-date, and reproducible studies assessing the prevalence and characteristics of OA. We address this need using oaDOI, an open online service that determines OA status for 67 million articles. We use three samples, each of 100,000 articles, to investigate OA in three populations: (1) all journal articles assigned a Crossref DOI, (2) recent journal articles indexed in Web of Science, and (3) articles viewed by users of Unpaywall, an open-source browser extension that lets users find OA articles using oaDOI. We estimate that at least 28% of the scholarly literature is OA (19M in total) and that this proportion is growing, driven particularly by growth in Gold and Hybrid. The most recent year analyzed (2015) also has the highest percentage of OA (45%). Because of this growth, and the fact that readers disproportionately access newer articles, we find that Unpaywall users encounter OA quite frequently: 47% of articles they view are OA. Notably, the most common mechanism for OA is not Gold, Green, or Hybrid OA, but rather an under-discussed category we dub Bronze: articles made free-to-read on the publisher website, without an explicit Open license. We also examine the citation impact of OA articles, corroborating the so-called open-access citation advantage: accounting for age and discipline, OA articles receive 18% more citations than average, an effect driven primarily by Green and Hybrid OA. We encourage further research using the free oaDOI service, as a way to inform OA policy and practice.

## INTRODUCTION

The movement to provide open access (OA) to all research literature is now over fifteen years old. In the last few years, several developments suggest that after years of work, a sea change is imminent in OA. First, funding institutions are increasingly mandating OA publishing for grantees. In addition to the US National Institutes of Health, which mandated OA in 2008 (https://publicaccess.nih.gov/index.htm), the Bill and Melinda Gates Foundation (http://www.gatesfoundation.org/How-We-Work/General-Information/Open-Access-Policy), the European Commission (http://ec.europa.eu/research/participants/data/ref/h2020/grants_manual/hi/oa_pilot/h2020-hi-oa-pilot-guide_en.pdf), the US National Science Foundation (https://www.nsf.gov/pubs/2015/nsf15052/nsf15052.pdf), and the Wellcome Trust (https://wellcome.ac.uk/press-release/wellcome-trust-strengthens-its-open-access-policy), among others, have made OA diffusion mandatory for grantees. Second, several tools have sprung up to build value atop the growing OA corpus. These include discovery platforms like ScienceOpen and 1Science, and browser-based extensions like the Open Access Button, Canary Haz, and Unpaywall. Third, Sci-Hub (a website offering pirate access to full text articles) has built an enormous user base, provoking newly intense conversation around the ethics and efficiency of paywall publishing (*Bohannon, 2016*; *Greshake, 2017*). Academic social networks like ResearchGate and Academia.edu now offer authors an increasingly popular but controversial solution to author self-archiving (*Björk, 2016a*; *Björk, 2016b*). Finally, the increasing growth in the cost of toll-access subscriptions, particularly via so-called ''Big Deals'' from publishers, has begun to force libraries and other institutions to initiate large-scale subscription cancellations; recent examples include Caltech, the University of Maryland, University of Konstanz, Université de Montréal, and the national system of Peru (*Université de Montréal, 2017*; *Schiermeier & Mega, 2017*; *Anderson, 2017a*; *Université Konstanz, 2014*). As the toll-access status quo becomes increasingly unaffordable, institutions are looking to OA as part of their ''Plan B'' to maintain access to essential literature (*Antelman, 2017*).

Open access is thus provoking a new surge of investment, controversy, and relevance across a wide group of stakeholders. We may be approaching a moment of great importance in the development of OA, and indeed of the scholarly communication system. However, despite the recent flurry of development and conversation around OA, there is a need for large-scale, high-quality data on the growth and composition of the OA literature itself. In particular, there is a need for a data-driven ''state of OA'' overview that is (a) large-scale, (b) up-to-date, and (c) reproducible. This paper attempts to provide such an overview, using a new open web service called oaDOI that finds links to legally-available OA scholarly articles.[1] Building on data provided by the oaDOI service, we answer the following questions:

1. What percentage of the scholarly literature is OA, and how does this percentage vary according to publisher, discipline, and publication year?
2. Are OA papers more highly-cited than their toll-access counterparts?

The next section provides a brief review of the background literature for this paper, followed by a description of the datasets and methods used, as well as details on the

[1] In the interest of full disclosure, it should be noted that two of the authors of the paper are the co-founders of Impactstory, the non-profit organization that developed oaDOI.

definition and accuracy of the oaDOI categorization. Results are then presented, in turn, for each research question, and are followed by a general discussion and conclusions.

## LITERATURE REVIEW

Fifteen years of OA research have produced a significant body of literature, a complete review of which falls outside the scope of this paper (for recent, in-depth reviews, see *Tennant et al. (2016)* and *McKiernan et al. (2016)*. Here we instead briefly review three major topics from the OA literature: defining OA and its subtypes, assessing the prevalence of OA, and examining the relative citation impact of OA.

Despite the large literature on OA, the term itself remains "somewhat fluid" (Antelman, 2004), making an authoritative definition challenging. The most influential definition of OA comes from the 2002 Budapest Open Access Initiative (BOAI), and defines OA as making content both *free to read* and *free to reuse*, requiring the opportunity of OA users to "crawl (articles) for indexing, pass them as data to software, or use them for any other lawful purpose." In practice, the BOAI definition is roughly equivalent to the popular "CC-BY" Creative Commons license (*Creative Commons, 2018*). However, a number of other sources prefer a less strict definition, requiring only that OA "makes the research literature free to read online" (*Willinsky, 2003*), or that it is "digital, online, [and] free of charge." (*Matsubayashi et al., 2009*). Others have suggested it is more valuable to think of OA as a spectrum (*Chen & Olijhoek, 2016*).

Researchers have identified a number of subtypes of OA; some of these have near-universal support, while others remain quite controversial. We will not attempt a comprehensive list of these, but instead note several that have particular relevance for the current study.

- Libre OA (*Suber, 2008*): extends user's rights to read and also to reuse literature for purposes like automated crawling, archiving, or other purposes. The Libre OA definition is quite similar to the BOAI definition of OA.
- Gratis OA (*Suber, 2008*): in contrast to Libre, Gratis extends *only* rights to read articles.
- Gold OA: articles are published in an "OA journal," a journal in which all articles are open directly on the journal website. In practice, OA journals are most often defined by their inclusion in the Directory of Open Access Journals (DOAJ) (*Archambault et al., 2014*; *Gargouri et al., 2012*).
- Green OA: Green articles are published in a toll-access journal, but self-archived in an OA archive. These "OA archives" are either disciplinary repositories like ArXiv, or "institutional repositories (IRs) operated by universities, and the archived articles may be either the published versions, or electronic preprints (*Harnad et al., 2008*). Most Green OA articles do not meet the BOAI definition of OA since they do not extend reuse rights (making them Gratis OA).
- Hybrid OA: articles are published in a subscription journal but are immediately free to read under an open license, in exchange for an an article processing charge (APC) paid by authors (*Walker & Soichi, 1998*; *Laakso & Björk, 2013*).

- Delayed OA: articles are published in a subscription journal, but are made free to read after an embargo period (*Willinsky, 2009*; *Laakso & Björk, 2013*).
- Academic Social Networks (ASN): Articles are shared by authors using commercial online social networks like ResearchGate and Academia.edu. While some include these in definitions of OA (*Archambault et al., 2013*; *Björk, 2016b*), others argue that content shared on ASNs is not OA at all. Unlike Green OA repositories, ASNs do not check for copyright compliance, and therefore as much as half their content is illegally posted and hosted (*Jamali, 2017*). This raises concerns over the persistence of content, since, as was the case in October 2017, publishers can and do issue large-scale takedown notices to ASN ordering the removal of infringing content (*Chawla, 2017*). Others have raised questions about the sustainability and ethics of ASN services themselves (*Fortney & Gonder, 2015*). Due to these concerns, and inconsistent support from the literature, we exclude ASN-hosted content from our definition of OA.[2]
- "Black OA": Articles shared on illegal pirate sites, primarily Sci-Hub and LibGen. Although (*Björk, 2017*) labels these articles as a subtype of OA, the literature has nearly no support for including Sci-Hub articles in definitions of OA. Given this, we exclude Sci-Hub and LibGen content from our definition of OA.

Based on the consensus (and in some cases, lack of consensus) around these definitions and subtypes, we will use the following definition of OA in the remainder of this paper: **OA articles are free to read online, either on the publisher website or in an OA repository.**

## Prevalence of OA

Many studies have estimated what proportion of the literature is available OA, including *Björk et al. (2010)*, *Laakso et al. (2011)*, *Laakso & Björk (2012)*, *Gargouri et al. (2012)*, *Archambault et al. (2013)*, *Archambault et al. (2014)* and *Chen (2013)*. We are not aware of any studies since 2014. The most recent two analyses estimate that more than 50% of papers are now freely available online, when one includes both OA and ASNs. *Archambault et al. (2014)*, the most comprehensive study to date, estimates that of papers published between 2011 and 2013, 12% of articles could be retrieved from the journal website, 6% from repositories, and 31% by other mechanisms (including ASNs). *Archambault et al. (2014)* also found that the availability of papers published between 1996 and 2011 increased by 4% between April 2013 and April 2014, noting that "backfilling" is a significant contributor to green OA. Their discipline-level analysis confirmed the findings of other studies, that the proportion of OA is relatively high in biomedical research and math, while notably low in engineering, chemistry, and the humanities.

This *Archambault et al. (2014)* study is of particular interest because it used automated web scraping to find and identify OA content; most earlier efforts have relied on laborious manual checking of the DOAJ, publisher webpages, Google, and/or Google Scholar (though see *Hajjem, Harnad & Gingras (2006)* for a notable early exception). By using automated methods, Archambault et al. were able to sample hundreds of thousands of articles, greatly improving statistical power and supporting more nuanced inferences. Moreover, by creating a system that indexes OA content, they address a major concern in the world of OA research; as *Laakso et al. (2011)* observes: "A major challenge for research...has been the

[2]Repositories that were included are those covered by the Bielefeld Academic Search Engine (BASE) in May 2017. A full listing of repositories can be found on their website at: https://www.base-search.net/about/en/about_sources_date.php?menu=2&submenu=1

lack of comprehensive indexing for both OA journals and their articles." The automated system of *Archambault et al. (2014)* is very accurate—it only misclassifies a paper as OA 1% of the time, and finds about 75% of all OA papers that exist online, as per *Archambault et al. (2016)*. However, the algorithm is not able to distinguish Gold from Hybrid OA. More problematically for researchers, the database used in the study is not open online for use in follow-up research. Instead, the data has since been used to build the commercial subscription-access database 1science (http://www.1science.com/oanumbr.html).

### The open access citation advantage

Several dozen studies have compared the citation counts of OA articles and toll-access articles. Most of these have reported higher citation counts for OA, suggesting a so-called "open access citation advantage" (OACA); several annotated bibliographies have been created to track this literature (*SPARC Europe, 2015*; *Wagner, 2010*; *Tennant, 2017*). The OACA is not universally supported. Many studies supporting the OACA have been criticised on methodological grounds (*Davis & Walters, 2011*), and an investigation using the randomized-control trial method failed to find evidence of an OACA (*Davis, 2011*). However, recent investigations using robust methods have continued to observe an OACA. For instance, *McCabe & Snyder (2014)* used a complex statistical model to remove confounding effects of author selection (authors may selectively publish their higher-impact work as OA), reporting a small but meaningful 8% OACA. *Archambault et al. (2014)* describe a 40% OACA in a massive sample of over one million articles using field-normalized citation rates. *Ottaviani (2016)* used a natural experiment as articles (not selected by authors) emerged from embargoes to become OA, and reports a 19% OACA excluding the author self-selection bias for older articles outside their prime citation years.

## METHODS

### OA determination
#### Classifications

We classify publications into two categories, OA and Closed. As described above, we define OA as *free to read online, either on the publisher website or in an OA repository*; all articles not meeting this definition were defined as Closed. We further divide the OA literature into one of four exclusive subcategories, resulting in a five-category classification system for articles:

- **Gold**: Published in an open-access journal that is indexed by the DOAJ.
- **Green**: Toll-access on the publisher page, but there is a free copy in an OA repository.
- **Hybrid**: Free under an open license in a toll-access journal.
- **Bronze**: Free to read on the publisher page, but without an clearly identifiable license.
- **Closed**: All other articles, including those shared only on an ASN or in Sci-Hub.

These categories are largely consistent with their use throughout the OA literature, although a few clarifications are useful. First, we (like many other OA studies) do not include ASN-hosted content as OA. Second, categories are exclusive, and publisher-hosted content takes precedence over self-archived content. This means that if an article is posted

in both a Gold journal and an OA repository, we would classify it as Gold, not Green. Put another way, publisher-hosted content can "shadow" archived articles that would otherwise be Green. This definition of Green ("available in a repository but *not* available from the publisher") is often used in the OA literature (including by Steven Harnad, the coiner of the Green and Gold terms *Harnad et al., 2008*), but this usage is not unanimous. Some studies allow a given article to be *both* Gold and Green; compared to these, our classification system does undercount Green. Hybrid articles share properties with Gold articles (both are free to read and are licensed for re-use), but differ in the venue of publication (i.e., Hybrid articles are published in journals not considered open access by the DOAJ) and in that Hybrid articles are not necessarily immediately available (i.e., they may only be freely available after an embargo). We also add a novel subcategory, Bronze. Bronze shares attributes of Gold and Hybrid; like both, Bronze OA articles are publisher-hosted. Unlike Gold OA, Bronze articles are not published in journals considered open access in the DOAJ. Unlike Hybrid, Bronze articles carry no license information. Although this lack of identifiable license may not be intentional, without an identifiable license, the articles are free to read but do not allow extended reuse rights beyond reading. It is also not clear if Bronze articles are temporarily or permanently available to read for free.

Finally, we should add that, although our categories of choice reflect the OA literature, they do not necessarily reflect the more complex reality of scholarly publishing today. Organizations like SciELO and Redalyc in Latin America have been acting simultaneously as publishers and repositories and many of the articles found on their site do not fall neatly into the above categories (*Packer, 2010*).

### The oaDOI system

We assigned the categories above by calling the oaDOI service with a DOI for each item. The oaDOI returns a link to a legally-available OA version of the article, when one is available (https://oadoi.org/). It contains records for all 88 million Crossref DOIs.[3] The oaDOI service crawls, aggregates, normalizes, and verifies data from many sources including PMC (https://www.ncbi.nlm.nih.gov/pmc/), BASE (https://www.base-search.net/about/en/), DOAJ (https://doaj.org/), and thousands of institutional repositories and publishers. The oaDOI system offers a fast, free API with no rate-limits, allowing it to support a variety of other services and tools. At the time of writing, oaDOI processes approximately 500,000 requests daily–roughly twice the daily uses of Sci-Hub[4] (*Bohannon, 2016*; *Himmelstein et al., 2017*). The majority of this volume comes from around 700 academic libraries, who use oaDOI to help readers find articles where the library has no subscription access, addressing the discoverability problem (*Chen, 2013*). The oaDOI service also powers the Unpaywall browser extension, which helps readers to find legal OA copies of paywalled articles as they browse; Unpaywall currently has over 80,000 active users. The oaDOI codebase is open source, and the service is free and open via an open API.

### Accuracy of oaDOI

To assess the accuracy of our automated OA determination, a random subsample of 500 articles were chosen from our main "Crossref-DOI" sample, described below. We manually searched the internet for each article in our subsample to determine if the

[3]DOIs are short, unique identifiers for scholarly papers. Crossref is a nonprofit that helps a the DOI system, and is by far the largest supplier of academic DOIs in academia.

[4]Based on a Sci-Hub dataset released in 2016 (the most recent data available).

**Table 1  Accuracy of the prototype version of the oaDOI service used in this study.**

|  | oaDOI reports Open | oaDOI reports Closed | Manual count Total (ground truth) |
|---|---|---|---|
| Open | 144 | 43 | 187 |
| Closed | 5 | 308 | 313 |
| Total | 149 | 351 | 500 |

paper was freely available on the publisher's website, or on another website, such as an institutional repository, an academic social networking site, or on a personal webpage. DOIs were resolved by appending the DOI to "https://doi.org/". If the full text was available through that link, articles were marked as being freely available from the publisher's site. If articles required a subscription, the title of the article was entered into Google Scholar (GS) and into Google to find alternative versions (i.e., preprints or archived copies). If the fulltext was found on any publisher page or OA repository, these were marked as being freely available from an archive. If the only available open copy was hosted on an academic social network (like Academia.edu or ResearchGate), this was noted but for the sake of the study these were *not* counted as any category of OA, and were instead added to the "Closed" category;

The performance of oaDOI is summarized below, compared to these manual accuracy checks. The complete dataset behind this summary is available in supplementary information. Using this data we calculated the recall and precision of the system. "Recall" asks the question, "when an article is open, how often does oaDOI correctly identify it as open?" The recall of the service is 77.0%, meaning that 77% of the truly open articles are correctly identified as open by oaDOI. "Precision" asks the question, "When oaDOI says an article is open, how often is it correct?" The precision of the system is 96.6%, meaning that 96.6% of the time that oaDOI reports an article is open, it really is open.

These results can be roughly compared to the recall of 86.4% and precision of 99.1% reported by *Archambault et al. (2014)* for their automated system. Their accuracy estimate was also calculated based on a sample of 500 data points, giving each estimate a margin of error of ±4.5 percentage points. The Archambault study used a narrower date window for their sample (starting in 1996, versus our Crossref-DOI sample which was not time restricted), resulting in a more homogeneous task, which may partially explain their somewhat better performance.

The oaDOI service is optimized for high precision, rather than high recall. The very high precision of oaDOI means that any estimates derived from the database can be considered a *conservative* estimate of the actual percentage of open access in the literature. That is, we can safely assume that when oaDOI reports a certain percentage of open access, the real percentage is *at least* that high—and almost certainly higher given that recall was less than perfect. Put another way, oaDOI delivers very few false positives (where it mistakenly calls an article open), but a relatively high number of false negatives (where it mistakenly calls an article closed) (Table 1). Future improvements to the system are planned that will improve recall while keeping precision high.
**Table 2** Summary of samples used in this study.

| Sample name | Sample size | Population sampled | Purpose | Population size |
|---|---|---|---|---|
| Crossref-DOIs | 100,000 | All journal articles with Crossref DOIs, all years. | Estimate percentage of the literature that is OA. | 66,560,153 |
| WoS-DOIs | 100,000 | All citable WoS articles with DOIs, 2009–2015. | Estimate citation impact of recent OA papers, and also OA prevalence by discipline. | 8,083,613 |
| Unpaywall-DOIs | 100,000 | All articles accessed by Unpaywall users over a 1-week period in 2017. | Estimate percentage of OA experienced by users of the Unpaywall extension. | 213,323 |

## Study samples

Three samples of DOI-assigned scholarly resources are summarized in Table 2 and described further below.

### *Crossref sample*

The first sample, "Crossref-DOIs," is a random sample of 100,000 journal articles with Crossref DOIs, across all publication years. There are approximately 88 million Crossref DOIs in total as of May 2017. In order to exclude books, datasets, and other non-article content, we sampled only items whose "type" was listed as "journal-article" in the Crossref API metadata; there are 66 million of these. To verify the accuracy of Crossref metadata, we manually checked 150 items assigned to type "journal-article," and determined that 93% were indeed journal articles; the remaining 7% were mostly journal front-matter such as tables of content or instructions to authors.

The purpose of this sample is to roughly proxy the scholarly literature as a whole. As such, it has strengths and weaknesses. One weakness is that although Crossref includes information on citation counts and discipline categorization, we found these to be quite incomplete, and therefore not useful for the present study. Another is that researchers in the scientometrics and OA fields have largely relied on other indexes, particularly Scopus and Web of Science (WoS), to represent the literature as a whole; this makes our results more difficult to compare to previous work. Finally, DOIs are known to be less frequently assigned by publishers in certain disciplines (like humanities; *Gorraiz et al., 2016*), in certain geographic regions (particularly the developing world), and among older articles (*Boudry & Chartron, 2017*); consequently, these segments will be underrepresented in our sample. This said, Scopus and WoS are also known to underrepresent important segments of the literature (*Mongeon & Paul-Hus, 2016*), and so this failing is not limited to Crossref. Moreover, the Crossref sample has important advantages of its own over other indexes. While no sample of the scholarly literature will be complete in every regard, the Crossref index is more expansive than other sources: in July 2017 there were 67 million journal articles indexed in Crossref compared to 30 million in Scopus (https://www.elsevier.com/solutions/scopus/content). Also, Crossref has the advantage of being entirely free and open to use, while Scopus and WoS are subscription-access databases; this allows the study data to also be free and open, promoting replication and

reuse of our results in further research. However, we did turn to the subscription-access WoS in order to answer questions about the discipline and citation counts of OA articles, since Crossref data is lacking in these areas.

### WoS sample

The second sample, "WoS-DOIs", is a random sample of 100,000 journal articles with DOIs that are indexed by Web of Science. The sample was drawn from a local version of the WoS database at the Observatoire des sciences et des technologies (OST) at the Université du Québec à Montréal. Only articles that WoS defines as "citable items" are included in the sample; this excludes non-peer reviewed content such as editorial material and news items. This sample is restricted to articles published between 2009 and 2015, due to DOI availability constraints. The sample of 100,000 articles is randomly drawn from a population of 8 million articles and reviews with a DOI in WoS published between 2009 and 2015 as of May 2017.

Because the WoS sample is restricted to certain publication years, due to availability of DOIs in the WoS database, this sample is unsuitable for estimating the proportion of the total literature that is OA. However, it is more useful than the Crossref sample in some ways: the WoS sample included accurate discipline information for each article (described below), and also citation counts. Therefore we use the WoS sample to assess OA prevalence by discipline and also the citation impact of recent OA papers. We do not encourage comparisons between the OA percentages in the WoS sample and the Crossref sample, because of large differences in the sampling frames.

Documents in the WoS-DOIs sample were classified using the National Science Foundation (NSF) journal classification system. This system assigns every journal exactly one "discipline" (a high-level categorization) and exactly one "specialty" (a finer-grained categorization). Because this is a journal-level classification, all articles from a given journal are assigned the same discipline and specialty as the journal. A downside of this approach is that the system classifies multidisciplinary journals (e.g., Nature, PNAS, PLOS ONE) as "biomedical research", despite their publishing many articles from other fields.[5] In these cases, we used a ground-up, article-by-article classification approach. Each article published in a list of multidisciplinary journals was assigned to the NSF specialty which appeared most frequently in its own reference list. In other words, papers published in multidisciplinary journals were classified at the article level (instead of at the journal level) to the subject area which they cite most frequently.[6]

We assess the relative impact of open and closed articles, using citations as an indicator of their scholarly impact. There are several properties of articles, however, that can confound this kind of comparison. Chief among these are the article's discipline (some fields are much more cited than others) and its age (older articles have had more time to gather citations). In order to address this, we computed a normalized expected number of citations for each article, based on its age and its NSF specialty, by comparing it to the average citations for similar articles.[7]

Using this approach, each article receives an average relative citation (ARC). An ARC of 1.0 indicates that a document was cited according to expectations based on documents

---

[5]These journals were identified by selecting journals with over a one thousand articles per year from those classified in the general "biomedical research" category. The full list of journals meeting these criteria were: PLOS ONE, Nature, Science, Scientific Reports, PNAS, Nature Communication, PeerJ, and Science Advances.

[6]Ties between frequently cited specialties were resolved randomly; that is, if a paper cites exactly the same amount of papers from two NSF specialties, it was assigned to one of the two at random

[7]Citations were normalized using the population of WoS articles and reviews with a DOI.

published in the same year and NSF specialty, while an ARC above or below 1.0 indicates that the citation impact was above or below world average, respectively. Using these field-normalized citation rates, citation impact can be compared across scientific disciplines as well as across years. We can also compute mean ARCs for groups of articles, like "all open articles" or "all closed articles", allowing us to compare normalized impact between these two groups. Analyzing results on the level of NSF disciplines, data is not shown for the Humanities ($n = 1,091$) and Arts ($n = 164$), because they are underrepresented both in the Web of Science and in terms of DOI coverage.

### Unpaywall sample

The third sample, "Unpaywall-DOIs", is a random sample of 100,000 articles accessed by users of the free, open-source Unpaywall browser extension, gathered over a one-week time window. We collected IP addresses and DOI requests made to the oaDOI service through the Unpaywall browser extension during the week of June 5–June 11, 2017. In that time period there were 374,703 total accesses, 213,323 unique DOIs, and 42,894 unique IP addresses gathered in total, from which 100,000 unique DOIs were randomly sampled.

This sample was used to assess the prevalence of OA experienced by users of the Unpaywall extension (since Unpaywall uses oaDOI data to find OA). It is a convenience sample of what articles people are interested in reading, and thereby lets us roughly estimate the percent of this literature that is OA. The sample has serious limitations, however: we don't know the demographics of Unpaywall users, and we are aware of a bias towards users from the US (as determined by the IP addresses). As such, we cannot accurately generalize the results by education level, discipline, or purpose in reading the scholarly literature.

## RESULTS

### RQ1. What percent of the literature is open access?
### How much of the literature is OA?

We found 27.9% (95% CI [27.6–28.2]) of all DOI-assigned journal articles are OA, using the Crossref-DOI sample. Based on this, we estimate there are 18.6 million OA articles with Crossref DOIs (95% CI [18.4–18.8]). This is the total population of OA articles that can be identified and accessed by oaDOI. Given our finding (described in Methods above) that the oaDOI service finds 77% of OA compared to manual searches, we can further estimate that an additional 3.5 million articles are OA but not detectable by this version of oaDOI.

People reading the literature using the Unpaywall browser extension encounter a significantly higher proportion of OA: we found that 47.0% (95% CI [46.7–47.3]) of the Unpaywall-accessed sample is open access. The main reason for this is article age: since this sample is based on the behavior of actual readers, it is disproportionately comprised of recent articles. In fact, half the accessed articles were published in the last 2 years. Recent articles are much more likely to be OA than their older counterparts (see Results 'How does Open Access vary by year of publication?' below).

### What types of Open Access are most common?

The proportion of OA by subtype is relatively similar across the samples, as shown in Fig. 1 and Table 3. Green OA represents a relatively small percentage of OA articles in all

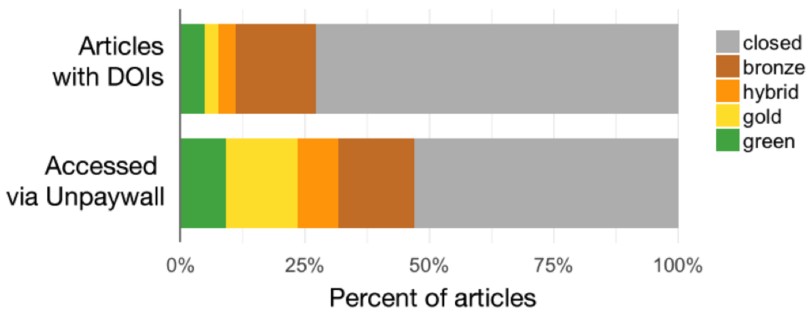

**Figure 1  Percent of articles by OA status, Crossref-DOIs sample vs Unpaywall-DOIs sample.**

**Table 3  Percent of the literature that is OA, by type, in three samples of 100,000 journal articles, with 95% confidence intervals.**

| Access type | Crossref-DOI All journal articles with Crossref DOIs, all years. ("Articles with DOIs" in Fig. 1) | | WoS-DOIs All citable WoS articles with DOIs, 2009–2015 | | Unpaywall-DOIs All articles accessed by Unpaywall users over a 1-week period in 2017 | |
|---|---|---|---|---|---|---|
| | Estimate | 95% CI | Estimate | 95% CI | Estimate | 95% CI |
| OA (all types) | 27.9% | 27.6–28.2 | 36.1% | 36.0–36.2 | 47.0% | 46.7–47.3 |
| Bronze OA | 16.2% | 16.0–16.5 | 12.9% | 12.6–13.2 | 15.3% | 15.0–15.6 |
| Hybrid OA | 3.6% | 3.3–3.9 | 4.3% | 4.0–4.6 | 8.3% | 8.0–8.6 |
| Gold OA | 3.2% | 2.9–3.5 | 7.4% | 7.1–7.7 | 14.3% | 14.0–14.6 |
| Green OA | 4.8% | 4.5–5.1 | 11.5% | 11.2–11.8 | 9.1% | 8.8–9.4 |
| Closed | 72.0% | 71.8–72.4 | 63.9% | 63.8–64.0 | 53.0% | 52.7–53.3 |

three samples. This is partly because self-archived articles are only counted as Green where there is no publisher-hosted option available; that is, Green OA is sometimes "shadowed" by Gold, Bronze, or Hybrid articles. Bronze is the most common OA subtype in all the samples, which is particularly interesting given that few studies have highlighted its role. We manually inspected a small sample of Bronze articles in order to understand this subcategory more; we found that while many Bronze articles were Delayed OA from toll-access publishers, nearly half were hosted on journals that published 100% of content as free-to-read but were *not* listed on the DOAJ and did not formally license content (using CC-BY or any other license). Such journals might be better described as "Dark Gold" or "Hidden Gold" than Bronze. A more complete examination of Bronze falls outside the scope of this study, and therefore further investigation will be undertaken in future work.

### How does Open Access vary by year of publication?

Figure 2 presents the number (Fig. 2A) and proportion (Fig. 2B) of papers by access category and publication date. Articles published in the last 20 years are increasingly OA, and this trend shows no sign of slowing. More recent articles are more likely to be OA, with the most recent year examined also containing the most OA: 44.7% of 2015 articles are OA

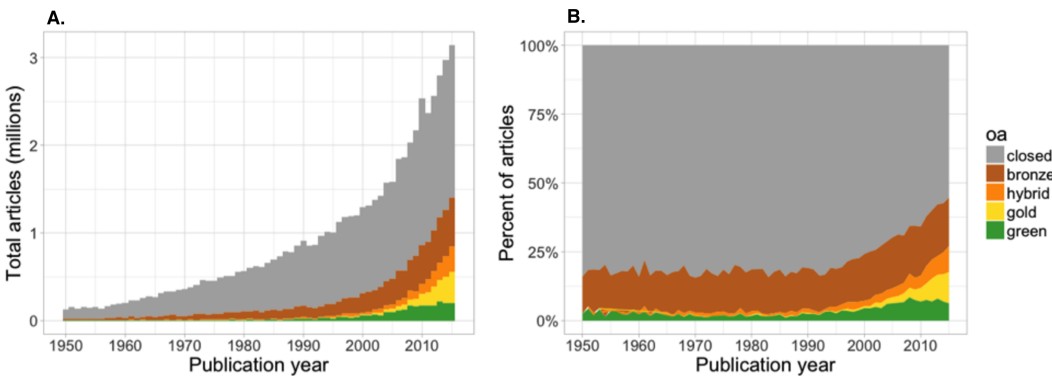

**Figure 2** Number of articles (A) and proportion of articles (B) with OA copies, estimated based on a random sample of 100,000 articles with Crossref DOIs.

(95% CI [43.3–46.2%]), including 17.6% Bronze (95% CI [16.2–19.1]), 9.4% Hybrid (95% CI [8.0–10.9]), 11.3% Gold (95% CI [9.9–12.8]), and 6.3% Green (95% CI [4.9–7.8]). Well over one million OA papers were published in 2015. This growth trend has largely been driven by dramatic growth in Gold and Hybrid OA since the year 2000. However, more than 20% of papers published before the digital age are also freely available. The majority of these older OA papers are Bronze, and based on their age they are probably more precisely Delayed OA, although additional investigation will be required to confirm this. Bronze OA remains remarkably constant as a proportion of the literature for all publication years examined.

The number and proportion of Green papers must be interpreted with particular caution, due to several factors. First, unlike publisher-hosted OA (Gold, Bronze, and Hybrid), the date when the Green article *became open* is generally different from the date the article was *first published*. Authors often self-archive articles years after (or before, in the case of preprints) their original publication, leading to so-called "backfilling" of Green stocks (*Archambault et al., 2014*). Consequently, the graph cannot show the growth of Green OA over time; this would require longitudinal analysis over several years, and so is outside the scope of this analysis. Instead it shows the number and proportion of Green OA by publication year of the article. Second, many articles cannot be legally self-archived until a certain number of months after publication; this embargoing likely influences the apparent plateau in Green shown in Fig. 2. Finally, as noted earlier, many self-archived articles would otherwise be Green except for being "shadowed" by a Gold, Bronze, or Hybrid of the same article elsewhere. For more detail on the growth of shadowed Green OA, see Figs. SA2 and SA3.

### How does Open Access vary by publisher?

We analyzed a subset of the Crossref-DOIs sample by publisher (as listed on the Crossref metadata record) to understand how the extent and types of OA are common across publishers for recent publications (between 2009 and 2015). As we can see in Fig. 3A, the largest publishers by volume publish the most OA articles by volume, led by Elsevier. As
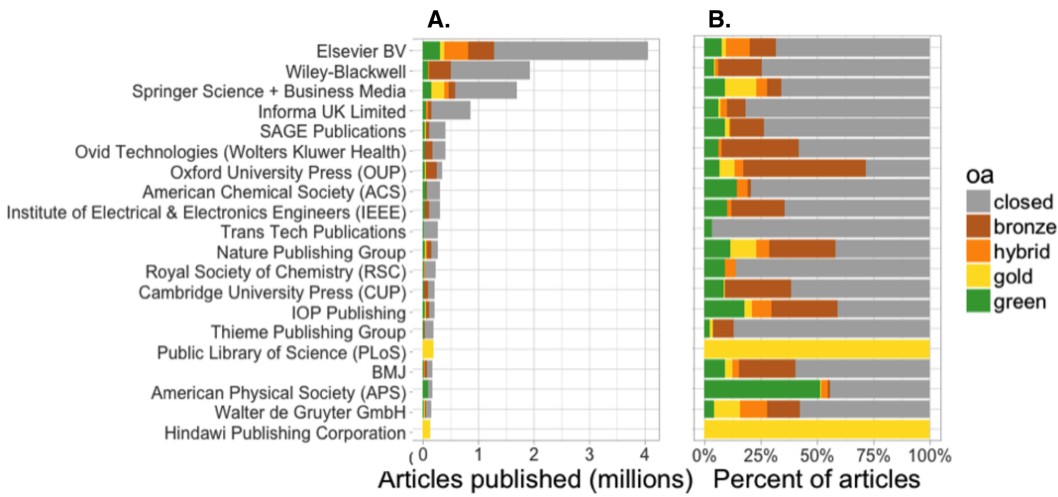

**Figure 3** Number (A) and proportion (B) of articles with OA copies, by publisher, for the 20 most prolific publishers. Based on sample of 27,894 Crossref DOI-assigned articles published between 2009–2015.

a proportion of all articles published (Fig. 3B), however, PLOS and Hindawi distinguish themselves as being the only publishers in the top 20 with 100% OA. More than half of the papers published by Oxford University Press, Nature Publishing Group, IOP Publishing, and the American Physical Society (APS) are freely available online. In the case of APS this is largely driven by content available through repositories such as arXiv (for more details on repositories, see Fig. SA1).

### How does Open Access vary across disciplines?

We used the WoS-DOIs sample to examine OA prevalence differences by discipline, because of the easy availability of discipline metadata in the WoS index. Figure 4 displays our results. More than half of the publications are freely available in biomedical research and mathematics, while in chemistry and engineering & technology less than 20% of the papers are freely available. Figure 4 also highlights the popularity of Green OA in disciplines like physics and mathematics, where more than one fifth of papers are available only through online repositories (mainly arXiv). Hybrid articles are particularly prevalent in mathematics (9.4%), biomedical research (8.1%) and clinical medicine (6.3%), while authors in biomedical research (15.3%), health (11.7%), mathematics (11.2%) and clinical medicine (10.3%) often publish in Gold journals.

Large variations can also be observed on the more detailed level of NSF specialties (Fig. SA5). At more than 80% of OA articles, astronomy & astrophysics (87%), fertility (86%), tropical medicine (84%), and embryology (83%) were the specialties where access to literature was the most open. At the other end of the spectrum are pharmacy (7%), inorganic & nuclear chemistry (7%), and chemical engineering (9%), where publications were hidden behind a paywall for more than 90% of papers. More detail on these and other NSF specialties can be seen in Fig. SA1.

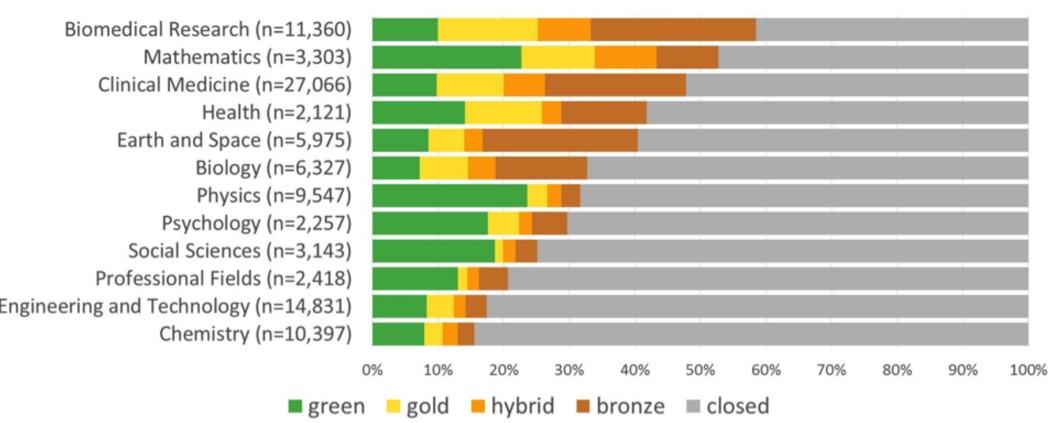

**Figure 4** Percentage of different access types of a random sample of WoS articles and reviews with a DOI published between 2009 and 2015 per NSF discipline (excluding Arts and Humanities).

### RQ2. What is the scholarly impact of open access?

Comparing the average relative citation impact of different access categories, the OACA is corroborated: Papers hidden behind a paywall were cited 10% below world average (ARC = 0.90), while those that are freely available obtain, on average, 18% more citations than what is expected (ARC = 1.18). However, citation impact differs between the different manners in which papers are made available for free: those that are only available as Green OA (ARC = 1.33) and Hybrid OA papers (ARC = 1.31) are cited the most with an impact of more than 30% above expectations, those available as Bronze are cited 22% above world average, while papers published as Gold OA obtain an ARC of 0.83. This constitutes an average relative citation impact of 17% below world average and 9% below that of articles hidden behind a paywall. Figure 5 below describes these findings.

These trends vary over time, however, as shown in Fig. 6. While the ARC of closed access papers remains below world average throughout the period studied, it increased from .86 in 2009 to .93 over in 2014 and 2015. Meanwhile, when looking across all open types, the mean citation rate is consistently above the world average, fluctuating between 1.15 and 1.22. This fluctuation is guided by differences between the access types, with the impact of Hybrid OA papers increasing over the time period. While Green OA papers' mean citation rate remain relatively stable, the highest impact, for 2015, is obtained by Bronze and Hybrid. The only form of open for which mean impact has decreased steadily over time is Gold. The results for more recent years are only based on a short citation window, however, and results might change over the next years as citations accumulate.

### DISCUSSION AND CONCLUSION

Access to scholarly literature is at the heart of current debates in the research community. Research funders are increasingly mandating OA dissemination to their grantees while, at the same time, the growth in toll-access subscriptions costs have prompted more and more university libraries to cancel subscriptions. In this context, several tools have been developed to provide access–both legally and illegally–to scholarly literature. Using data

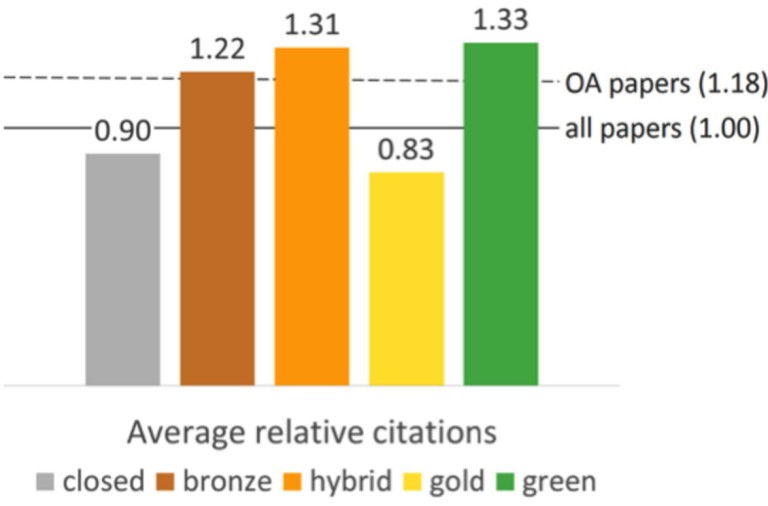

**Figure 5** Average relative citations of different access types of a random sample of WoS articles and reviews with a DOI published between 2009 and 2015.

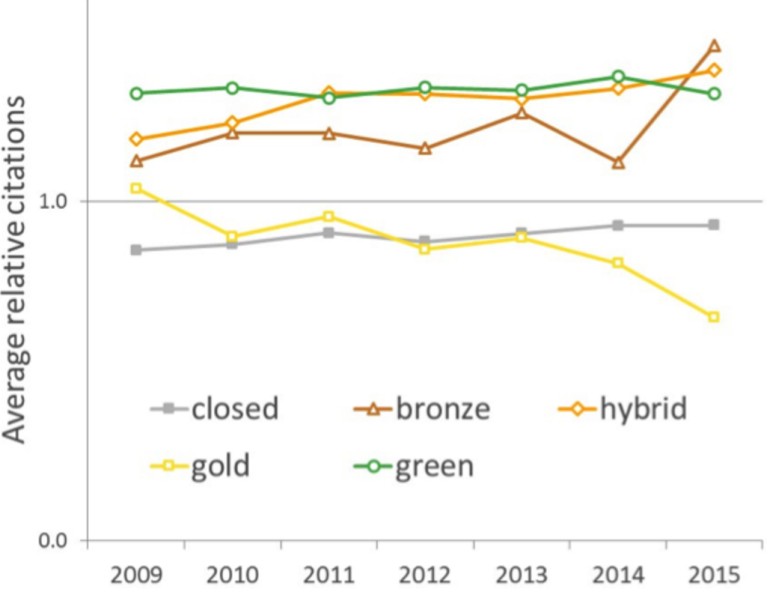

**Figure 6** Percentage and impact of different access types of a random sample of WoS articles and reviews with a DOI, by year of publication.

from one of these tools (oaDOI), this paper addresses two broad research questions: what percent of the literature is OA and how does it vary by type of OA, and what is the mean scholarly impact of papers diffused through this form. Three large samples were used, to assess different aspects of OA patterns: (1) 100,000 articles that have a Crossref DOIs, which allows us to assess the relative proportion of OA across all existing literature; (2) 100,000 WoS-indexed journals articles that have a DOI, which allows us to assess the

scholarly impact of OA and non OA papers; (3) 100,000 articles accessed by users through the Unpaywall browser extension, which lets us assess the proportion of OA papers found by users of this free tool.

We found that 28% of all journal articles are freely available online (Crossref-DOI sample). Encouragingly for proponents of OA, this proportion has been growing steadily over the last 20 years, driven particularly by growth in Gold and Hybrid. Articles from 2015, the most recent year examined, had the highest proportion OA (45%), as well as the largest absolute number of OA articles published in a single year. This disproportionate level of OA in recent years, combined with readers' preference for more recent articles, leads to a felicitous situation for readers: the proportion of OA they *experience* as they browse and search is better than the overall percentage of OA across the literature as a whole. Users of the Unpaywall browser extension, which gives individual readers access to the oaDOI service, encounter OA articles nearly half (47%) of the time. The effect almost certainly extends beyond Unpaywall users; one may assume readers in general also favor newer articles, and therefore benefit from the growth of Gold, Bronze, and Hybrid OA among recent papers, even without using Unpaywall. More studies of readership data from other sources would be useful to quantify this further.

Interestingly, we found that the majority of OA articles are Bronze–hosted on publisher websites, either without a license at all or without an open license. This is surprisingly high given that Bronze is relatively little-discussed in the OA literature, and suggests that this OA category deserves further attention from the OA community. In particular, Bronze OA may be significant in a policy context, since, unlike other publisher-hosted OA, Bronze articles do not extend any reuse rights beyond reading, making them Gratis OA. Much more research is needed into the characteristics of Bronze OA. How many Bronze articles are licensed openly, but do not make their license available? Is Bronze disproportionately non-peer-reviewed content? How much of Bronze OA is also Delayed OA? How much Bronze is Promotional, and how transient is the free-to-read status of this content? How many Bronze articles are published in "hidden gold" journals that are not listed in the DOAJ? Why are these journals not defining an explicit license for their content, and are there effective ways to encourage this? These and other questions are outside the scope of this study but may provide fruitful insights for future OA research and policy.

Only about 7% of the literature overall (and 17% of the OA literature) is Green. This is may at first seem disappointing, given years of advocacy focused on Green OA as well as ongoing growth in the number of Green OA mandates (*Björk et al., 2014*). However, the full context of Green OA provides reasons for optimism. First, many papers are archived in repositories but are not counted as Green in this analysis because they are also available on the publisher site as Hybrid, Gold, or Bronze versions. These "shadowed Green" copies provide a useful safety net that preserves access in cases where publishers rescind it (as could potentially happen with Delayed OA and other Bronze articles). Further research is needed to determine the prevalence of shadowed Green OA in various disciplines. Second, the phenomenon of "backfilling" (authors self-archiving content published across all years, not just the current one) means that although the percentage graph of Green OA does not show the same year-over-year slope as Gold or Hybrid, the line itself may be rising

across *all* years as authors gradually self-archive papers from years or even decades ago. This assumption is supported by results reported by *Archambault et al. (2016)*. Finally, the relatively low proportion of green OA encouragingly leaves room for continued growth. While most journals published by major publishers (Elsevier, Wiley, Springer, etc.) allow for self-archiving, research shows that only a small proportion of papers from these publishers actually are self-archived in OA repositories; for example, *Smith et al. (in press)* report using a sample of Global Health Research papers that only 39% of them made use of available self-archiving rights.

Our results confirm the Open Access Citation Advantage found by other studies: open articles receive 18% more citations than otherwise expected. While at least some of this boost is likely due to the fact that more access allows more people to read and hence cite articles they otherwise would not, causation is difficult to establish and there are many possible confounders. Most discussed is the so-called "selection bias postulate", (*Craig et al., 2007*) which suggests that authors choose only their most impactful work to make OA. The current study does not examine the cause or directionality of correlation, but does find that it exists in a very large sample that is relatively representative of the literature as a whole. Funder requirements may also play a role in the observed citation advantage: high-profile funders are more likely to have an OA publishing requirement; at the same time, well funded studies are independently more likely to receive more citations than poorly funded studies (*Berg, 2010*). Interestingly, Gold articles are actually cited *less*, likely due to an increase in the number of newer and smaller OA journals. Some of these journals are from regions of the world not historically indexed by WoS, are published in languages other than English, or might be considered to be less prestigious because they have not had time to become established or accumulate citations (*Archambault et al., 2013*). On the flip side, the citation disadvantage of Gold OA is likely also affected by the continued growth of so-called 'mega journals' such as PLOS ONE (*PLOS, 2018*). Whatever the reason, the lower impact of Gold means the overall citation advantage is strongly driven by Green, Hybrid, and Bronze content. In sum, while several factors can affect the observed differences in citation rates, and causation remains difficult to establish, the fact remains that scholars are much more likely to read and cite papers to which they have access than those that they cannot obtain. Hopefully the existence of a free, open index of OA content will help support further research into the OACA question.

The relatively high percentage of OA found in this study, particularly among readers of the free Unpaywall extension, has important potential implications for academic libraries. Increasingly, these libraries are under pressure to meet growing prices of "Big Deal" subscription packages, and the once-unthinkable outcome of canceling these Big Deals is becoming an increasingly realistic option. In this environment, knowing that around half of the literature of interest is available without any subscription may tip the scales toward cancellation for some institutions–particularly given that this percentage seems to be growing steadily. Indeed, the Université de Montréal's cancellation of their Taylor & Francis subscription package (*Université de Montréal, 2017*) is particularly interesting, given that their cancellation announcement directly pointed faculty to Unpaywall and other tools to help them access OA content. This may seem a radical suggestion, but cancellation

of subscription journals has long been part of the universal OA roadmap (*Anderson, 2017b*). Even when the percentage of OA is not enough to support outright cancellation, it may be enough to negotiate better subscription rates by supporting calculation of "OA-adjusted Cost Per Access" (*Antelman, 2017*). However, much more study is needed to see how OA availability varies across journals and Big Deal packages, along with praxis-oriented work building OA analysis tools that help librarians make cancellation choices.

This study has several important limitations. Our dataset only includes journal articles with DOIs, which means that disciplines and geographical areas which rely more heavily on conference papers or articles without DOIs are underrepresented. Our Crossref sample includes about 7% journal "front matter" that the journal has assigned a DOI and Crossref labelled "journal article" but is actually a page describing the journal Editorial Board or similar. Our Bronze OA category includes articles published in OA journals which aren't indexed in DOAJ; future work must identify these OA journals and classify such articles as Gold. As discussed in our definition of OA, when finding open copies we ignored free-to-read articles from academic social networks like ResearchGate and Academia.edu. The oaDOI system has some coverage of articles published on personal web pages, but this is quite limited compared to web-scale indexes like Google. The oaDOI system includes thousands of institutional and subject repositories, but there are some repositories that it misses. Our accuracy checks suggest that oaDOI, and therefore this study, are probably overlooking around 23% of OA otherwise discoverable using web searches, meaning that estimates in reported in this paper undercount OA by approximately 30%. Finally, our approach did not detect *when* articles were deposited into repositories. Because repositories are often backfilled with content that has been published many years ago, this study does not measure any increase/decrease in prevalence of Green OA over time, but only the proportion of Green OA by article publication date at the moment of data collection.

In addition to the empirical results obtained, this paper clearly shows the potential of the oaDOI service for future research. The freely available oaDOI service provides scholars with the basis for assessing and monitoring the development of access to scholarly literature on a large scale, as well as the factors that affect it. For instance, our results show that the percentage of the literature available as OA is growing, and that articles diffused through this form are generally more cited than closed access articles. Several factors are likely to contribute to these trends; however, those remain poorly understood. Combined with other datasets–such as the WoS, Scopus, or Crossref–oaDOI allows one to assess at a large-scale the effects of various mandates on deposit rates, or to track the development of documents' accessibility to determine, for example, when authors self-archive, or the sustainability of the promotional OA category. Aggregated at the level of journals and publishing platforms, these data can also provide librarians with indicators to help inform subscription cancellations and mitigate their effects. The application of the oaDOI algorithm on a large scale also allows for more complete analysis of the OA citation advantage across fields and time. As in *Gargouri et al. (2010)*, confounding factors could be mitigated by using article-level metadata to identify article pairs published in the same journal issue, on the same topic or published by the same authors at the same time. We hope that other scholars will dig deeper in those data to better understand OA

dissemination and the factors that drive it. This is of utmost importance for the future of scholarly communication.

## ACKNOWLEDGEMENTS

The authors would like to thank Dorothea Salo, Kristin Antelman, and John Sack for extensive and valuable comments on a draft of this article. The author order of JP and HP was determined by coin flip, as is their custom.

### Funding
The authors received no funding for this work.

### Competing Interests
Heather Piwowar and Jason Priem are founders of Impactstory, a non-profit company which makes Unpaywall, oaDOI, and other tools to improve scholarly communication.

### Author Contributions
- Heather Piwowar, Jason Priem and Stefanie Haustein conceived and designed the experiments, performed the experiments, analyzed the data, contributed reagents/materials/analysis tools, wrote the paper, prepared figures and/or tables, reviewed drafts of the paper.
- Vincent Larivière conceived and designed the experiments, performed the experiments, analyzed the data, contributed reagents/materials/analysis tools, wrote the paper, reviewed drafts of the paper.
- Juan Pablo Alperin conceived and designed the experiments, performed the experiments, analyzed the data, wrote the paper, reviewed drafts of the paper.
- Lisa Matthias performed the experiments, analyzed the data, reviewed drafts of the paper.
- Bree Norlander analyzed the data, wrote the paper, reviewed drafts of the paper.
- Ashley Farley wrote the paper, reviewed drafts of the paper.
- Jevin West reviewed drafts of the paper.

### Data Availability
Zenodo: http://doi.org/10.5281/zenodo.837902.

The datasets behind the analysis in this paper are openly available at http://dx.doi.org/10.5281/zenodo.837902 and the R statistics code can be found at https://github.com/Impactstory/oadoi-paper1. The oaDOI code is open source at https://github.com/impactstory/oadoi and information about accessing the oaDOI API and full dataset is at https://oadoi.org/api.

## Supplemental Information

Supplemental information for this article can be found online at http://dx.doi.org/10.7717/peerj.4375#supplemental-information.

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
