# Peer review of "The state of OA: a large-scale analysis of the prevalence and impact of Open Access articles"

_PeerJ, doi:10.7717/peerj.4375_

## Round 0.1 · original submission · Minor Revisions

The paper is, in general, clearly written and appears to adhere to PeerJ policies concerning data access and availability. However, the reviewers in general felt that there were some minor revisions that should be reviewed prior to publication.

Key among these was the change to the declaration in the competing interests section for two of the authors noted by Reviewer 1.

Other areas for review were in the citations and in typos noted by reviewers 2 and 3 and some suggestions for the data packaging by reviewer 3. Reviewer 3 also has some other suggestions for making a stronger case on several assertions and some clarifications for sampling procedures.

This is a solid paper and should be accepted with minor revisions.

Reviewer 1 ·

Basic reporting

This article is well and clearly written, and is characterized by (to the extent possible, given the subject matter) an admirable avoidance of jargon. It is well organized, with clear and logically-arranged sections. However, the authors do need to review their references: in at least one case, they cite an article in support of a point not addressed in that article, but rather in another one by the same author that is uncited in this piece. All data sets, including the source code for the tool under examination, are made publicly available.

Experimental design

Given that PeerJ "considers articles in the Biological Sciences, Medical Sciences, and Health Sciences" and "does not publish in the Physical Sciences, the Mathematical Sciences, the Social Sciences, or the Humanities (except where articles in those areas have clear applicability to the core areas of Biological, Medical or Health sciences)," it is not clear to me that this article is strictly in-scope. This strikes me as a piece that would fall within the category of the social sciences. However, the tool under discussion certainly has application to publications in biology, medicine, and the health sciences, so PeerJ's editors will have to decide whether the subject connection is strong enough to merit inclusion.

The research methodology described in the article strikes me as perfectly sound, and the authors' interpretation of the results seem reasonable to me.

Validity of the findings

The findings strike me as sound. Not everyone will necessarily be driven to the exact same conclusions from these data as those arrived at by the authors, but since the data are provided openly, there's no reason why those who disagree can't come to their own conclusions. This article adds a valuable and (I think) important piece of the puzzle in our ongoing quest to understand the OA publishing ecosystem.

The article's most significant problem is not related to its content, but rather to the fact that its authors have (unless I missed the disclosure somewhere) failed to disclose a serious conflict of interest. Nowhere in the article, that I can determine, is it revealed that both unpaywall and oaDOI are products of Impactstory -- the company of which the paper's first two authors are founders and principals. In fact, at several points in the paper (notably the final paragraph of its Discussion and Conclusion section, which states "this paper clearly shows the potential of the oaDOI service for future research"), it begins feeling like an infomercial for Impactstory's products. A particularly attentive reader might notice that the links to source data for the study lead to Impactstory domains, but this doesn't seem sufficient: instead, the lead authors should say right at the beginning of the paper that they are principals in and cofounders of the company that produces the tool the performance and potential of which are under examination in this article.

·

Basic reporting

Overall the article is well written and structured. There are some minor formatting issues with citations that should be reviewed and cleaned up (e.g. see first paragraph of lit review). I was surprised that Walt Crawford's extensive work on OA Journals wasn't referred to, but that may have been deemed out of scope.

The figures do not translate well to black and white and minor revision to make that more clear is recommended. The bar charts were incredibly difficult to understand without the color. This also made it unclear if the keys and the charts aligned.

A couple of typos scattered throughout (e.g. under definition of ASNs "we we"

Formatting of the definitions of OA vary a lot -- citations bounce all over the entries. Please update for consistency.

Raw data was available. Thank you for the very clear dictionary.

Experimental design

On Page 8 the authors mention using a broader date window than the Archambault sample -- was this only for the WOS portion? Please clarify.

On the top of page 11 where the authors describe the article assignment to discipline. It was unclear how the authors determined which journals were multidisciplinary and how articles were then further assigned. Was this by hand?

Validity of the findings

no comment

Additional comments

Overall I really enjoyed reading this paper. I made comments as I went through and found nearly all of these addressed later in the paper.

·

Basic reporting

A comprehensive description of oaDOI would be helpful earlier in the introduction. I was expecting more information about oaDOI in line 81, consider moving some of the language in line 230 to better frame the role of oaDOI in 'the state of OA'

Regarding the raw data and analysis available at the DOI provided, you might consider including a manifest with data and article DOIs as well as a more complete codebook or at least definitions of all variables and codes in dataset (ie, 999 for missing data). This is a valuable dataset for the community and is somewhat difficult to interpret given the current documentation.

In lines 307-308 the authors assert that Scopus and WoS are known to underrepresent segments of the literature. The article would benefit from a reference for this assertion.

There is a grammatical error on line 498 'this paper aimed at answering...'

Several references are formatted inconsistently, ie line 689

Experimental design

It would be helpful for the reader to clarify which ASNs were excluded from the analysis - all ASNs or only commercial ASNs? For example, would articles shared in the Humanities Commons be excluded?

Validity of the findings

It would be useful to address the potential for self-selection among users of the Unpaywall extension and the possibility that this might impact the Unpaywall sample.

In section 1.5 of the results, the conclusions are not well-stated. Consider a comparison of disciplines based on either the percentage of open papers or the percentage of gated papers. A combination of both is confusing for the reader.

In the discussion of Bronze OA (line 527), you state that it 'seems likely' for Bronze to be disproportionately non-peer-reviewed. Why does this seem likely?

Additional comments

I appreciated the opportunity to review this paper, this is a fascinating and very important analysis of open access scholarly literature. I am looking forward to seeing what additional research emerges from this, especially in the areas of citation advantage for Gold OA and the impact of Bronze OA.

·

Basic reporting

No comment.

Experimental design

No comment

Validity of the findings

No comment.

Additional comments

Wonderfully accessible and well framed for understanding by multiple audiences.
It would be interesting to see what could be done about parsing out the shadow of the green OA, since the availability of that content is likely more stable, and less subject to the whims of publishers than bronze (and some Gold, esp. in the hybrid case). Speaking of... the community would definitely welcome further research on the bronze OA concept, so I'm thrilled to see that you plan to develop that further.
Thank you for an open article about openness that uses open data!

---

## Round 0.2 · Minor Revisions

Overall the reviewers would like to thank the authors for taking our reviews under consideration and for the changes that were made to reflect these suggestions. Currently there is one other query made by the Anonymous reviewer that asks a specific question as follows:

"Line 69 contains a reference to a piece by Anderson ("The Forbidden Forecast") that isn't relevant to the point under discussion (current Big Deal cancelations). The authors may have confused that piece with this one:

https://scholarlykitchen.sspnet.org/2017/05/01/wolf-finally-arrives-big-deal-cancelations-north-american-libraries/

Later in the paper the "Forbidden Forecast" piece is cited in the context of a more relevant point."

Please review the article link to ensure that this is the appropriate link for the article as intended. Once this has been reviewed we expect a quick turn around on publication."

Thank You,

Robert

Reviewer 1 ·

Basic reporting

Line 69 contains a reference to a piece by Anderson ("The Forbidden Forecast") that isn't relevant to the point under discussion (current Big Deal cancelations). The authors may have confused that piece with this one:

https://scholarlykitchen.sspnet.org/2017/05/01/wolf-finally-arrives-big-deal-cancelations-north-american-libraries/

Later in the paper the "Forbidden Forecast" piece is cited in the context of a more relevant point.

Experimental design

No change to previous comments

Validity of the findings

No change to previous comments

Additional comments

No change to previous comments

·

Basic reporting

No comment

Experimental design

No Comment

Validity of the findings

No Comment

Additional comments

Thank you for reviewing the edits suggested before and making updates.

·

Basic reporting

Thank you for your revisions and response. I maintain that the underlying data provided at http://doi.org/10.5281/zenodo.837902 could benefit from more documentation, including a manifest with data and article DOIs as well as a more complete codebook or at least definitions of all variables and codes in dataset. However, I believe that this revised manuscript meets the PeerJ criteria and am recommending that it be accepted as is.

Experimental design

No comment

Validity of the findings

No comment

---

## Round 0.3 · accepted · Accept

The paper is, in general, clearly written and adheres to PeerJ policies concerning data access and availability and offers contributions to the field of OA discussions for scholarly communications.

All reviewers thought that this was a solid paper and should be accepted.